# SARS-CoV-2 human challenge reveals biomarkers that discriminate early and late phases of respiratory viral infections

Blood transcriptional biomarkers of acute viral infections typically reflect type 1 interferon (IFN) signalling, but it is not known whether there are biological differences in their regulation that can be leveraged for distinct translational applications. We use high frequency sampling in the SARS-CoV-2 human challenge model to show induction of IFN-stimulated gene (ISG) expression with different temporal and cellular profiles. *MX1* gene expression correlates with a rapid and transient wave of ISG expression across all cell types, which may precede PCR detection of replicative infection. Another ISG, *IFI27*, shows a delayed but sustained response restricted to myeloid cells, attributable to gene and cell-specific epigenetic regulation. These findings are reproducible in experimental and naturally acquired infections with influenza, respiratory syncytial virus and rhinovirus. Blood *MX1* expression is superior to *IFI27* expression for diagnosis of early infection, as a correlate of viral load and for discrimination of virus culture positivity. Therefore, *MX1* expression offers potential to stratify patients for antiviral therapy or infection control interventions. Blood *IFI27* expression is superior to *MX1* expression for diagnostic accuracy across the time course of symptomatic infection and thereby, offers higher diagnostic yield for respiratory virus infections that incur a delay between transmission and testing.

Host response biomarkers of viral infection have multiple potential clinical applications. These include diagnostic triage tests to direct prioritisation of confirmatory laboratory investigations, to guide clinical management decisions with the aim of reducing unnecessary antibacterial prescribing, or trigger infection control measures and antiviral treatment. Attention has mostly focussed on biomarker discovery in whole blood samples that enable easy and technically consistent access. Genome-wide transcriptional profiling has emerged as the most common unbiased data-driven approach due to the maturity of technical and analytical workflows[1].

Numerous blood transcriptional signatures for host responses to viral infections have been identified in this way using case-control studies of natural infection or experimental viral challenge in humans, designed to discover the most parsimonious measurements that discriminate viral infections from healthy controls or other diseases. We previously tested the accuracy of such blood transcriptional signatures of viral infection, identified by systematic review, to detect incident SARS-CoV-2 infection[2]. We showed that the majority were highly correlated, and collectively driven by type 1 interferon (IFN) responses. Many, including single gene transcripts (such as that of *IFI27*) provided near perfect discrimination of PCR positive individuals compared to uninfected controls. In some, the transcriptional biomarkers identified infections before the first positive viral PCR in nasopharyngeal samples. The sensitivity of *IFI27* measurements was further leveraged to provide evidence for abortive infections associated with virus specific T cell responses without detection of the virus by PCR[3].

✉e-mail: m.noursadeghi@ucl.ac.uk

In observational studies of natural infection, it is not possible to synchronise the time course of exposure and replicative infection. This has precluded identification of temporally distinct host response biomarkers that may offer optimal solutions for different translational applications such as diagnostic triage or patient stratification for antiviral therapies. To address this limitation, we leveraged the first controlled human challenge model of SARS-CoV-2 infection, complemented with high frequency sampling to measure viral replication and host responses spanning the full time course of viral replication[4]. We updated our previous systematic review to undertake comprehensive head-to-head evaluation of all reported host transcriptional signatures of viral infection to date. We compared their ability to discriminate between groups of participants with and without evidence of replicative infection using whole blood samples stratified by time since experimental inoculation. For selected biomarkers, representative of differential host-responses over the time course, we evaluated associations with symptoms and viral load. We investigated their cellular source in single cell transcriptomic data, and the potential epigenetic mechanisms that may underpin their differential expression. We compared their measurement in blood and nasal swabs and explored the extent to which our findings were generalizable to other respiratory viruses in both experimental challenge and natural infection studies.

## Results

### Blood transcriptional signatures of viral infection

We updated our previous systematic review of the literature, to identify 26 blood transcriptional signatures associated with viral infection (Supplementary Fig. 1A, Supplementary Table 1, Supplementary Data 1)[5–29]. These included six single gene biomarkers. The remaining multigene signatures were made up of 2-47 constituent genes. The composition of these signatures was generally distinct, reflected by low Jaccard indices in a matrix of pairwise comparisons (Supplementary Fig. 1B).

### Viral infection outcomes in the SARS-CoV-2 controlled human challenge model

33 SARS-CoV-2 seronegative healthy volunteers subjected to nasal inoculation of a standardized dose of SARS-CoV-2 divided into two groups with (N = 17) and without (N = 16) evidence of sustained replicative infection from 2 days after challenge (Fig. 1). Although the individual viral load profiles were different in nose and throat swabs,

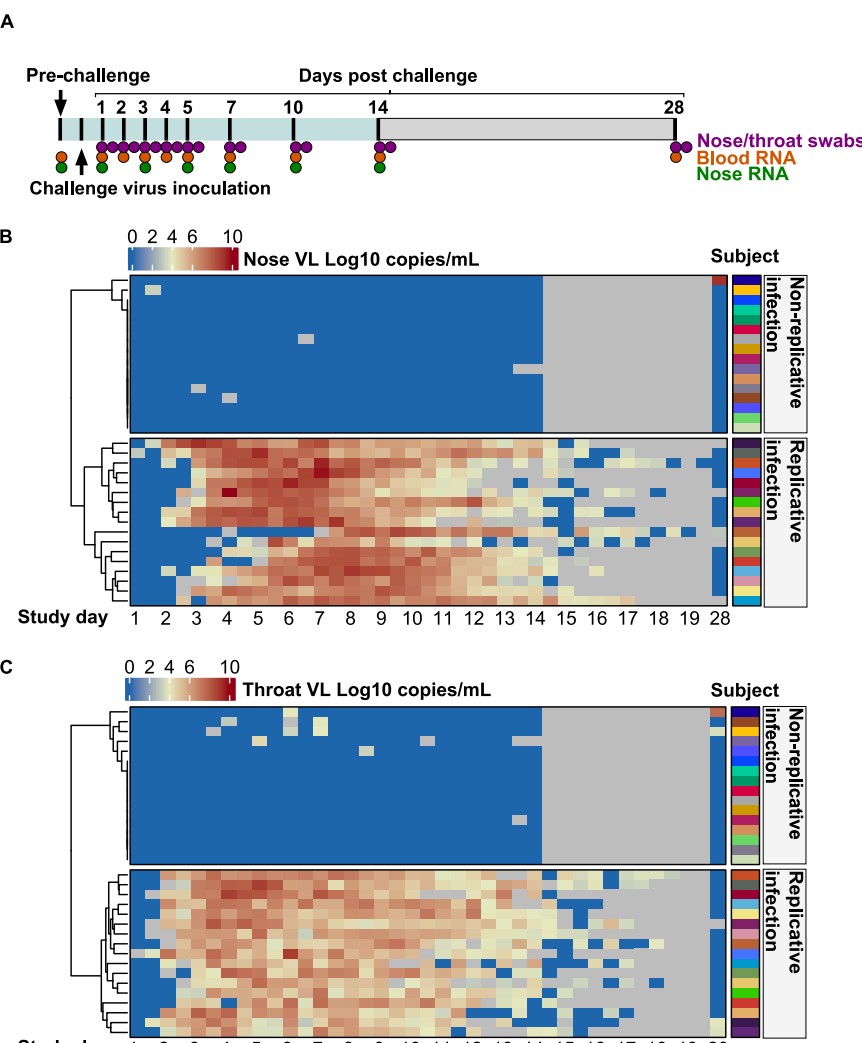

**Fig. 1 | SARS-CoV-2 PCR viral load in nose and throat swabs following virus challenge. A** Schematic overview of SARS-CoV-2 challenge of unvaccinated SARS-CoV-2 seronegative healthy adults showing time points for viral load measurements (nose and throat swabs), and host RNA sequencing (blood samples and nose swabs). Blue shading represents period of quarantine. **B** Quantitative viral load measurements by PCR from nose and (**C**) throat swabs per participant (rows) stratified by time point (columns) after virus challenge, and clustered (using complete linkage hierarchical clustering) into two groups of participants with (N = 17) and without (N = 16) evidence of replicative virus infection. Grey colour denotes unavailable data points.

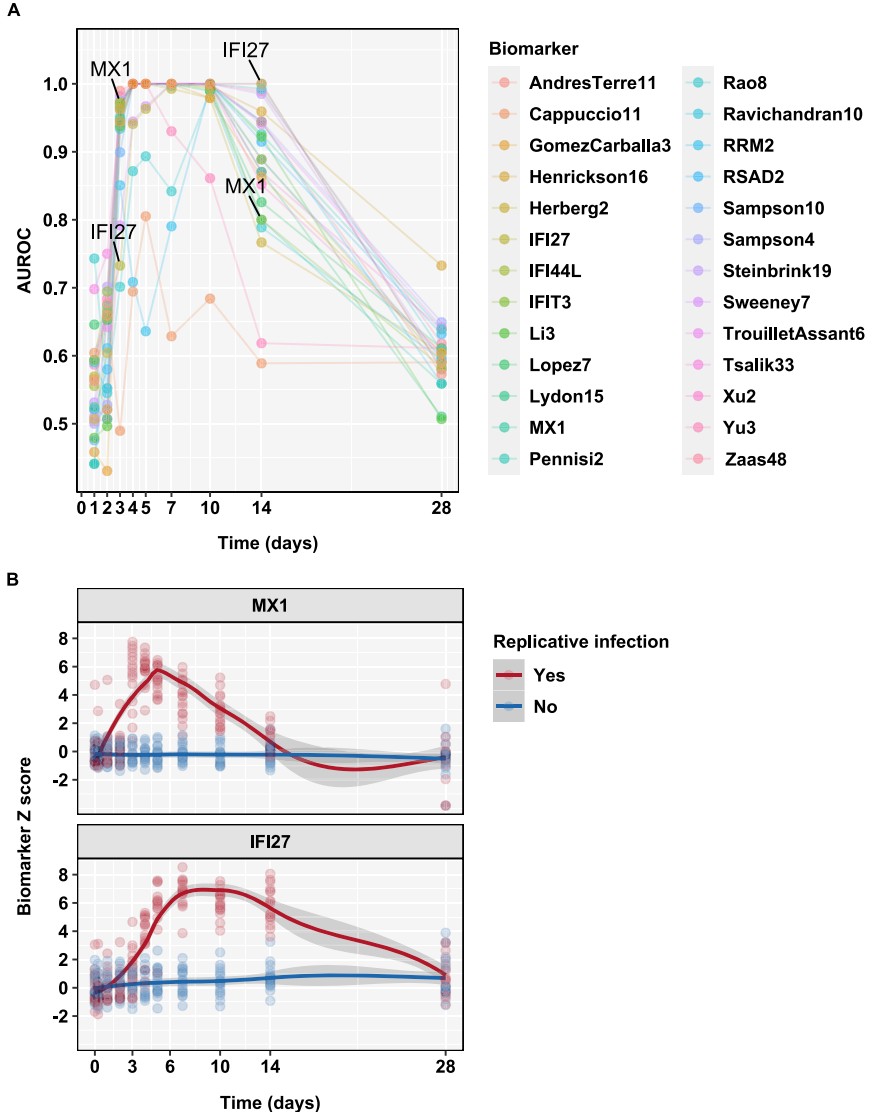

**Fig. 2 | Blood transcriptional discrimination of participants with and without replicative infection by time from SARS-CoV-2 challenge. A** Point estimates for area under the receiver operating characteristic curve (AUROC) stratified by blood transcriptional signature and time after virus challenge. **B** Individual (data points) and loess smoothed summary (line ±95% CI) for standardised blood transcript levels of *MX1* and *IFI27* in sequential time points after challenge, for participants with ($N = 17$) and without ($N = 16$) replicative viral infection.

both measurements segregated the same participants into two groups with and without replicative infection.

### Blood transcriptional biomarker discrimination of participants with and without sustained replicative SARS-CoV-2 infection

Blood transcriptional biomarker scores were calculated for each of the 26 signatures identified by systematic review, from RNA sequencing of whole blood samples at selected time points before and after viral inoculation (Fig. 2A). Across this time course, all the biomarkers showed a transient increase in expression (Supplementary Fig. 3) associated with replicative SARS-CoV-2 infection. We first ranked all biomarkers by their ability to discriminate between participants with and without replicative infection by area under the receiver operating characteristic curve (AUROC) across 14 days. We limited calculations to data from days 3, 7, 10 and 14, in order to achieve equal sampling frequency distribution across the time course of infection (Supplementary Fig. 3). Point estimates of the AUROCs ranged between 0.6–0.99. 22 of the 26 biomarkers with point estimates ranging 0.92–0.99 were statistically comparable with overlapping 95% confidence intervals, suggesting most biomarkers

were able to accurately discriminate participants with and without replicative infection.

### Identification of blood transcriptional biomarkers of early and late phases of SARS-CoV-2 infection

Next, we compared the AUROC of each signature stratified by time point. Most achieved near perfect discrimination of participants with and without replicative infection on days 4–10 (Supplementary Fig. 4A). We found greater variation in performance of each signature before and after this time interval, suggesting differential ability to identify early and late phases of viral infection. To investigate this hypothesis further, we focused on the single gene transcripts with highest AUROC on day 3 (*MX1*) and on day 14 (*IFI27*). On day 3, *MX1* achieved an AUROC of 0.97 (0.93-1) which reduced to 0.8 (0.64-0.96) by day 14. In contrast, *IFI27* achieved an AUROC of 0.73 (0.56-0.91) on day 3, increasing to 1 by day 14 (Fig. 2A). These findings reflected an early but transient increase in *MX1* expression and a comparatively delayed but sustained increase in *IFI27* expression (Fig. 2B). These distinct temporal profiles were comparable in male and female individuals (Supplementary Fig. 4B). A

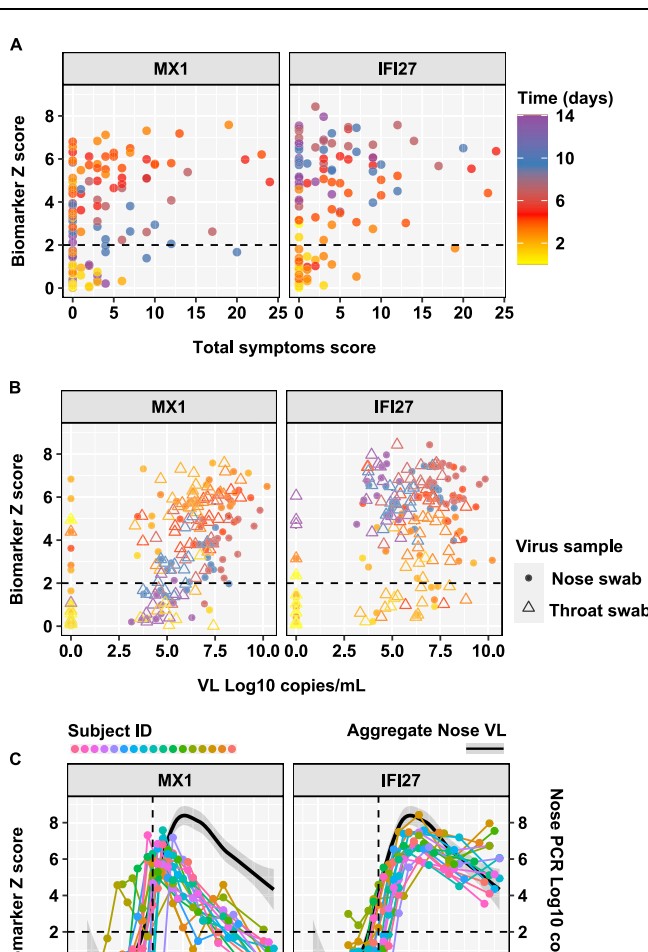

**Fig. 3 | Relationship between blood transcript levels of MX1 and IFI27 with symptoms and viral load by time from SARS-CoV-2 challenge.** Individual standardised blood transcript levels of *MX1* and *IFI27* against (**A**) symptom scores and (**B**) nose and throat viral loads in participants who developed replicative virus infection (*N* = 17), stratified by time after virus challenge, with dashed line to represent the threshold (Z > 2) for elevated standardised transcript levels. **C** Individual standardised blood transcript levels of *MX1* and *IFI27* (connected data points, left axis) and loess smoothed summary for nose viral load (black line ± 95% CI, right axis) in participants who developed replicative virus infection (*N* = 17), by time from virus detection in nose swabs > 4 Log10 copies/mL.

number of other single gene biomarkers (IFI44L, IFIT3 and RSAD2) were highly correlated to *MX1* and distinct from *IFI27* (Supplementary Fig. 4C).

### Relationship of MX1 and IFI27 expression in blood to symptoms and SARS-CoV-2 viral load

Most biomarker discovery and validation has focused on naturally acquired symptomatic viral infection. We and others have shown that host response biomarkers are able to detect asymptomatic infection[2,12]. Consistent with this, we found no correlation between blood transcriptional scores and prospective quantitation of daily symptom scores among individuals who developed replicative infection (Fig. 3A). We used a pre-specified threshold of Z score > 2 to indicate increased biomarker levels compared to baseline with 98% specificity. We found elevated biomarker scores at time points in which participants who experienced replicative infection were completely asymptomatic. This was more evident with *MX1*

measurements at early time points and with *IFI27* measurements at late time points.

In addition, we investigated the relationship between blood transcriptional signature scores and viral load stratified by time from inoculation in samples from individuals who developed replicative infection. Examples of elevated *MX1* and *IFI27* scores (Z > 2) were evident at time points with negative virus PCR in contemporary nose or throat swabs (Fig. 3B). Elevated *MX1* scores associated with negative virus PCR tests were more evident at early time points, and elevated *IFI27* scores associated with negative virus PCR tests were more evident at late time points. Importantly, *MX1* and *IFI27* scores in the normal range (< Z2) were also evident at time points with positive virus PCR tests in contemporary samples. False-negative biomarker results were more evident for *IFI27* at early time points, and for *MX1* at late time points. To underscore the differential temporal relationship of each biomarker with viral load, we examined longitudinal biomarker measurements per participant who developed replicative infection, indexed by time from first PCR detection of virus (> 4 Log10 copies/ mL) in nasal swabs, which we have recently reported to correlate best with viral emissions[30]. The rise in *MX1* scores was generally co-incident with PCR detection of the virus, and in some individuals evident before detection of virus by PCR. However, the *MX1* response generally peaked before the peak in viral load, suggesting that clearance of *MX1* transcript enrichment was faster than clearance of the virus. In contrast, *IFI27* scores increased after detection of the virus and remained elevated after viral load started to fall (Fig. 3C). Among time points in which at least one of these biomarkers was elevated (> Z2) to signify replicative infection the ratio of *MX1:IFI27* levels best correlated with time from virus challenge. In this analysis, the transition from a predominant *MX1* response to a predominant *IFI27* response occurred 5 days after virus challenge (Supplementary Fig. 5).

Both blood transcriptional biomarkers showed statistically significant correlation with viral load when including PCR negative time points (Supplementary Fig. 6A) consistent with the fact that they provided good discrimination of groups of participants with and without replicative infection. However, when restricting the analysis to time points with positive virus PCR tests, we found a significant correlation only to *MX1*, suggesting this biomarker provided better prediction of viral load than *IFI27*, which remained elevated at later time points (Fig. 3B, Supplementary Fig. 6B). Consistent with this observation, we also found that *MX1* provided a better biomarker of infectiousness than *IFI27*, by predicting positive viral culture in contemporary samples. Among individuals who developed replicative infection, blood *MX1* transcript levels discriminated virus culture positivity in nose or throat samples with AUROC 0.85 (0.79-0.92), significantly better than *IFI27* which achieved AUROC of 0.66 (0.57-0.75). In this analysis false positive *MX1* levels were limited to early time points, consistent with the observation that the rise in *MX1* levels can precede PCR detection of the virus (Fig. 4).

### Differential regulation of MX1 and IFI27 expression in blood

Both *MX1* and *IFI27* are widely recognised as interferon stimulated genes (ISG)[31,32]. To explore this relationship among participants in the replicative infection group, we compared *MX1* and *IFI27* levels with the average expression of a multigene signature ("STAT1 regulated module") that we had previously derived and validated as a measure of type 1 IFN bioactivity[33]. Both biomarkers showed a statistically significant correlation with the STAT1 module, but the relationship with *MX1* was stronger with near perfect correlation and covariance, suggesting that *IFI27* expression was subject to additional levels of transcriptional regulation (Fig. 5A, B). To obtain a deeper insight into the mechanisms of differential regulation of *MX1* and *IFI27*, we investigated their expression in our previously reported single cell RNA sequencing analysis of PBMC from a subset of participants with replicative infection in the present SARS-CoV-2 challenge study[34]. We found a clear

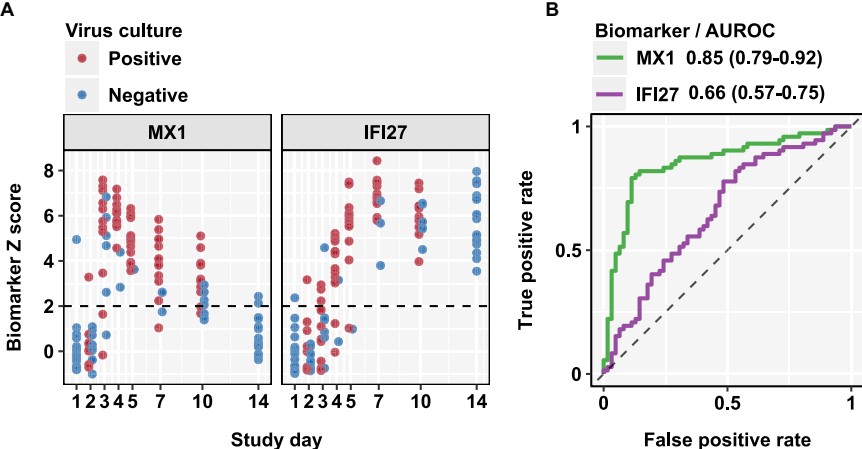

**Fig. 4 | Discrimination of virus culture positivity by blood transcript levels of MX1 and IFI27. A** Individual standardised blood transcript levels of *MX1* and *IFI27* at each time point for all individuals who develop replicative infection (*N* = 17), stratified by contemporary virus culture positivity in either nose or throat swabs, with dashed line to represent the threshold (Z > 2) for elevated transcript levels. **B** Area under the receiver operating characteristic curve (AUROC) discrimination of virus culture positivity by blood transcript levels of *MX1* and *IFI27* across all time points, showing AUROC point estimates and 95% confidence intervals.

increase of *MX1* expression in all major PBMC subsets in pooled day 3 data, and subsequent reduction by day 7. In contrast, increased expression of *IFI27* was almost exclusively restricted to myeloid cells (monocytes and conventional dendritic cells). Modest upregulation was evident at day 3, but then increased further at day 7 and day 10 before reducing again by day 14, although expression levels remained higher than baseline through to day 28 (Fig. 5C).

In published ATAC sequencing data[35], we tested the hypothesis that differential time and cellular distribution of *MX1* and *IFI27* expression reflected differential epigenetic regulation (chromatin accessibility) of *MX1* and *IFI27* loci in circulating immune cells. In datasets from unstimulated monocytes, CD4 T effector cells and B-cells from healthy individuals, we found evidence that the *MX1* locus contained areas of open chromatin (enrichment of sequencing peaks) close to the transcription start site and exon-1 (Supplementary Figs. 7A, B), which would enable rapid transcriptional upregulation of this gene across multiple cell types. In contrast, the *IFI27* locus contained little evidence of open chromatin (Supplementary Figs. 7A, B) in any of these cell types, and therefore inaccessible for rapid transcriptional upregulation. To evaluate subsequent epigenetic modifications following infection, we leveraged single cell ATACseq data from patients admitted to hospital with COVID-19[36]. Despite the sparsity in single cell data and relatively low coverage of the *IFI27* locus, in samples from patients with acute COVID-19, we found a higher number of IF27 sequencing reads in monocytes compared to all lymphocyte populations. This difference was less evident in data from convalescent patients (Supplementary Fig. 7C), and consistent with transient cell-type specific opening of the *IFI27* locus in established infection, providing a mechanistic basis for the temporal delay and cellular restriction of *IFI27* responses compared to *MX1*.

### Generalisable differential utility of blood MX1 and IFI27 transcriptional biomarkers in acute respiratory virus infection

In order to investigate whether the differential host responses represented by *MX1* and *IFI27* were generalisable to other acute respiratory viral infections, we investigated their expression profiles in collated data from previously reported influenza, respiratory syncytial virus, and rhinovirus human challenges among participants with evidence of infection following inoculation as per original study definitions[37]. In every case *MX1* upregulation in whole blood transcriptional profiles preceded that of *IFI27* (Fig. 6A). The data from these experiments were limited to ~6 days post-challenge and did not allow us to fully compare the temporal profiles of these biomarker measurements to the present SARS-CoV-2 challenge. Therefore, we undertook transcriptional profiling of blood samples from another recent H3N2 influenza human challenge model that included sampling beyond day 7[38]. This analysis also reproduced our findings in the SARS-CoV-2 challenge (Fig. 6B).

We further sought to extend the generalisability of our findings to natural infections. In a household contact study of index cases with COVID-19 spanning pre-alpha, and alpha-virus (B.1.1.7) pandemic waves in the UK[39], blood transcript levels of *MX1* and *IFI27* achieved equivalently good discrimination of contacts with and without prevalent SARS-CoV-2 infection at recruitment (day 0, AUROC 0.97, 0.92-1). This level of discrimination was maintained for *IFI27* in follow up samples 7 days later, but significantly reduced for *MX1*, consistent with earlier resolution of this biomarker (Fig. 7A). In a further data set from patients with unselected community acquired respiratory virus infections, we evaluated *MX1* and *IFI27* expression in whole blood transcriptional profiles of individuals with PCR confirmed respiratory virus infections within 48 hours of symptom onset, in four sequential samples on alternate days[40]. Compared to baseline (pre-infection) samples from the same individuals, increased levels of *MX1* expression (Z > 2) were largely confined to early time points day 0-2 after presentation within 4 days of symptom onset. Increased levels of *IFI27* expression (Z > 2) were evident over a longer time course including day 4-6 after presentation, up to 8 days after symptom onset (Fig. 7B, Supplementary Fig. 8A). Across all time points, *IFI27* measurements achieved statistically better AUROC than *MX1* measurements for discrimination of infection from baseline uninfected samples (Supplementary Fig. 8B). However, when the analysis was stratified by sample time point, *MX1* achieved the highest AUROC for discrimination of infected samples on the day of presentation (Supplementary Fig. 8C). The AUROC for *MX1* reduced significantly at each subsequent time point. The time point stratified analysis of *IFI27*, showed stable AUROC discrimination of infection. A combined biomarker signature, comprising the average expression of *MX1* and *IFI27* improved the AUROC discrimination at early time points compared to *IFI27* alone, and at late time points compared to *MX1* alone (Supplementary Fig. 8C).

### Comparison of host response biomarkers of acute respiratory virus infection in blood and nose samples

The potential to measure host response transcriptional signatures in samples from upper respiratory tract swabs has recently been reported[41,42]. We compared *MX1* and *IFI27* transcript measurements in samples from blood and nose swabs in the present SARS-CoV-2

challenge. Surface nose swabs only yielded adequate RNA for sequencing in 103 of 238 samples (43%), reflecting an inherent technical limitation in this approach. Nonetheless, for nose samples which did yield RNA sequencing data, we found clear evidence of *MX1* and *IFI27* responses in participants who developed a replicative infection. In comparison to blood measurements of these biomarkers, the signal strength in nose swab samples was weaker than in blood, the response in the nose was delayed in comparison to blood, and the differential time course for each biomarker evident in blood samples was lost in nose swab samples (Fig. 7C). These findings were replicated in blood and nasal mucosal curettage samples from the H3N2 influenza human challenge and indicate that in general, blood biomarker measurements are likely to provide better diagnostic discrimination for prevalent infection as well as better differentiation of early and late phases of infection, compared to nasal swabs (Fig. 7C).

## Discussion

We present a comprehensive evaluation of previously reported transcriptional signatures as host response biomarkers of viral infection in high frequency longitudinal blood and nose swab samples from the first SARS-CoV-2 human challenge experiment. We provide compelling evidence showing that single gene transcripts for *MX1* and *IFI27* in blood, discriminate temporally distinct phases of infection, and we

show that these findings are generalisable across a range of clinically important respiratory viruses in both experimental and naturally acquired infections. The earliest phase of replicative SARS-CoV-2 infection was associated with rapid upregulation of *MX1* transcripts in blood, which may precede PCR detection of the virus and correlated with PCR positive viral load measurements. In contrast, blood transcriptional upregulation of *IFI27* occurred after PCR detection of the virus. *IFI27* expression did not correlate with PCR positive viral load measurements and was sustained above baseline levels after viral clearance. Of note, transcriptional upregulation of both biomarkers was independent of symptoms.

Both *MX1* and *IFI27* are widely recognised as ISGs[31,32]. The *MX1* response closely reflected generalised type 1 ISG expression across all major cell types. We focused on *MX1* because it achieved the highest single gene point estimate AUROC for discriminating groups of individuals with and without replicative viral infection at the first time point at which any biomarker achieved significant discrimination. Alternative interferon inducible single gene biomarkers such as *IFI44L*, *IFIT3* and *RSAD2* provided statistically comparable discrimination at this time point, and are highly correlated to *MX1*. These biomarkers are likely to share the same mechanisms for transcriptional regulation, and offer the same utility as *MX1*. Delayed transcriptional upregulation of *IFI27* compared to other canonical ISGs has also been

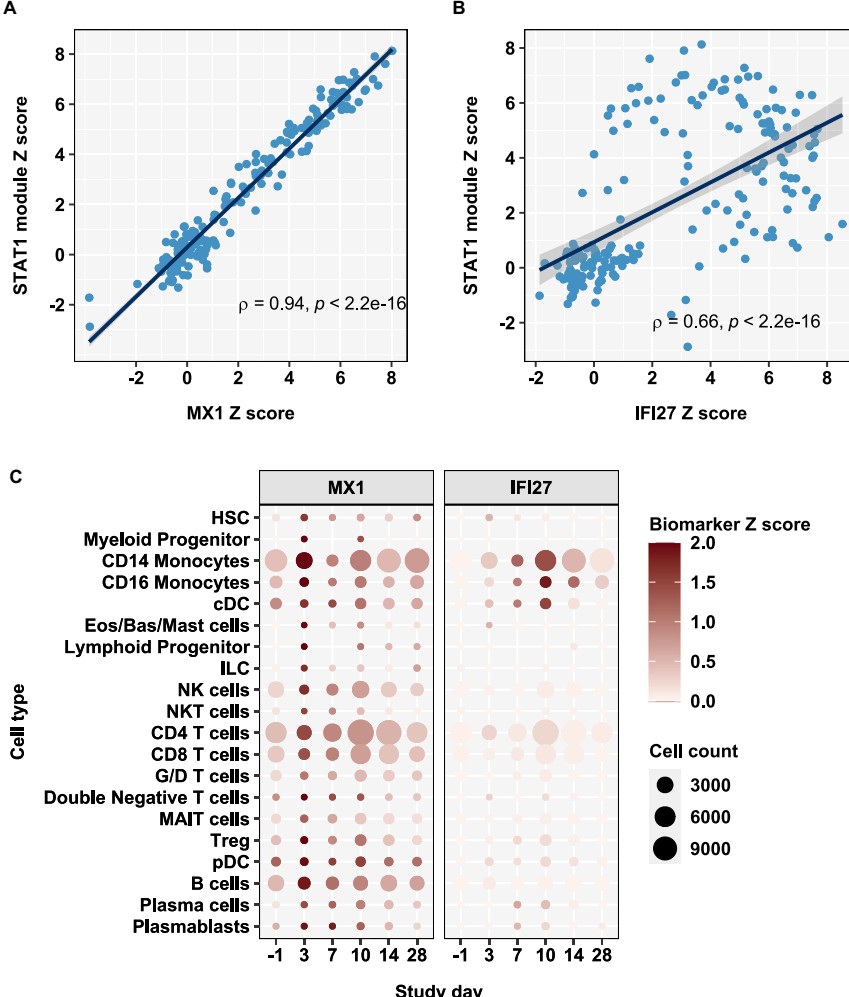

**Fig. 5 | Differential regulation of MX1 and IFI27 expression in blood.** Individual standardised blood transcriptional expression of a type 1 IFN stimulated gene signature (STAT1 module) by standardised blood transcriptional expression of (**A**) *MX1* and (**B**) *IFI27* at all time points in participants who developed replicative virus infection ($N = 17$), showing linear regression lines ±95% CI, 2-sided Spearmen correlation coefficients and *p* value. **C** Standardised blood transcriptional expression of *MX1* and *IFI27* stratified by cell type and time after virus challenge, in pooled data from participants who developed replicative virus infection ($N = 6$).

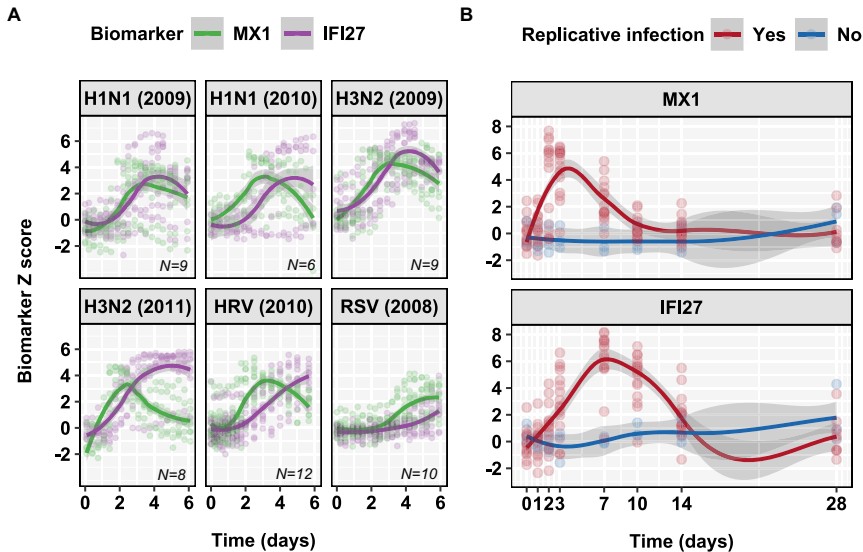

**Fig. 6 | Generalisable differences in temporal profiles of blood MX1 and IFI27 expression in diverse respiratory virus challenges.** Individual (data points) and loess smoothed summaries (lines ±95% CI) for (**A**) standardised blood transcript levels of *MX1* and *IFI27* over the first 6 days after challenge in selected human respiratory virus challenge models (GSE73072) among participants who develop replicative infection, and (**B**) over 14 days in an H3N2 influenza human challenge model, among participants with (*N* = 16) and without (*N* = 3) replicative infection.

reported following in vitro stimulation of cells with IFN[43,44]. In vivo, *IFI27* expression in blood samples was restricted to myeloid PBMC. We found evidence of differential epigenetic silencing of the *IFI27* locus compared to the *MX1* locus in resting PBMC, and cell type specific epigenetic modulation of this locus in monocytes during established COVID-19 infection. These data provide a mechanistic explanation for the differential temporal and cellular expression of the two biomarkers, namely that *IFI27* is epigenetically silenced in resting cells but becomes accessible for transcription in specific myeloid lineages during the evolving immune response to infection.

The differential temporal expression of *MX1* and *IFI27* in the SARS-CoV-2 challenge model was replicated in challenge experiments with multiple influenza strains, respiratory syncytial virus, or rhinovirus, and in data from household contacts with naturally acquired SARS-CoV-2 infection. Likewise, in unselected community acquired symptomatic respiratory virus infections, in which *MX1* measurements achieved high diagnostic accuracy for infections within 4 days of symptom onset, and *IFI27* measurements achieved higher diagnostic accuracy at later time points. The combination of both measurements provided highest diagnostic accuracy across all time points.

In SARS-CoV-2 and H3N2 Influenza challenge experiments, we found no evidence that measuring these biomarkers in nose swab samples offered any advantage to blood samples. Since *IFI27* expression in blood is largely restricted to myeloid cells, quantitatively lower Z scores for *IFI27* in nasal swabs may reflect lower sensitivity for host RNA biomarkers in superficial nasal swabs because these swabs capture fewer immune cells from sub-epithelial layers. *MX1* expression appears to be less restricted to specific cell types. Therefore, lower *MX1* Z scores suggest that nasal swabs may also be less sensitive because the sample provides lower sequencing read depth or the cells captured by nasal swabs do not upregulate IFN stimulated gene expression to the same extent as circulating immune cells. Unexpectedly, upregulation of these biomarkers in nose samples of individuals with replicative infection was also temporally delayed compared to the blood. This finding is also evident in our analysis of comparative single cell sequencing data from blood and nose swab samples. Whether it reflects faster transmission of IFN signalling to circulating blood cells, later onset of viral replication in the nose compared to the throat, or local suppression of IFN signalling by the virus in the nasal mucosa require future mechanistic investigation.

Comprehensive identification of reported blood transcriptional biomarkers of viral infection by systematic review, and their application in standardised SARS-CoV-2 and influenza human challenges with high frequency sampling are major strengths of this study, thus enabling identification of differential temporal profiles of *MX1* and *IFI27* responses. Single cell data from the SARS-CoV-2 challenge model, and analyses of publicly available data also allowed investigation of the mechanism for differential temporal profiles of *MX1* and *IFI27* responses, and to confirm reproducibility of our findings across a range of respiratory virus infections.

Our key conceptual advance is that different blood transcriptional biomarkers discriminate early and late phases of infection as a result of cell-specific epigenetic regulation and have important translational applications for respiratory virus infections. *MX1* transcripts may have specific diagnostic utility in early pre-symptomatic infection. Their correlation with viral load and infectiousness, represents the first evidence for a biomarker that can be used to identify patients most likely to benefit from antiviral treatments and infection control measures, such as quarantine/self-isolation. Finally, the delay between transmission and testing in naturally acquired infection means that overall *IFI27* transcripts achieve greater diagnostic accuracy than *MX1* transcripts as a diagnostic triage test for respiratory virus infection. To take full advantage of the differential temporal profiles, a combined approach including both *MX1* and *IFI27* (as averaged expression, or where a positive test for either gene triggers further confirmatory testing) may be the optimal approach to diagnostic triage.

Our conclusions are currently limited to data derived from individuals with non-severe infection. Therefore, future validation in hospitalised cohorts for whom the time of exposure can be estimated will be required to assess whether severe disease alters the temporal profiles of these biomarkers. In addition, whether anti-type 1 IFN antibodies which have emerged as a risk factor for severe COVID-19[45] may reduce the expression of IFN-inducible genes and thereby reduce the sensitivity of *MX1* and *IFI27* as diagnostic biomarkers will need to be addressed in future studies. Finally, we do not address the specificity of our findings for respiratory virus infections. Therefore, we have limited our discussion of potential translational applications to diagnostic triage tests to trigger confirmatory virological investigations, stratification of patients with confirmed viral infections for antiviral

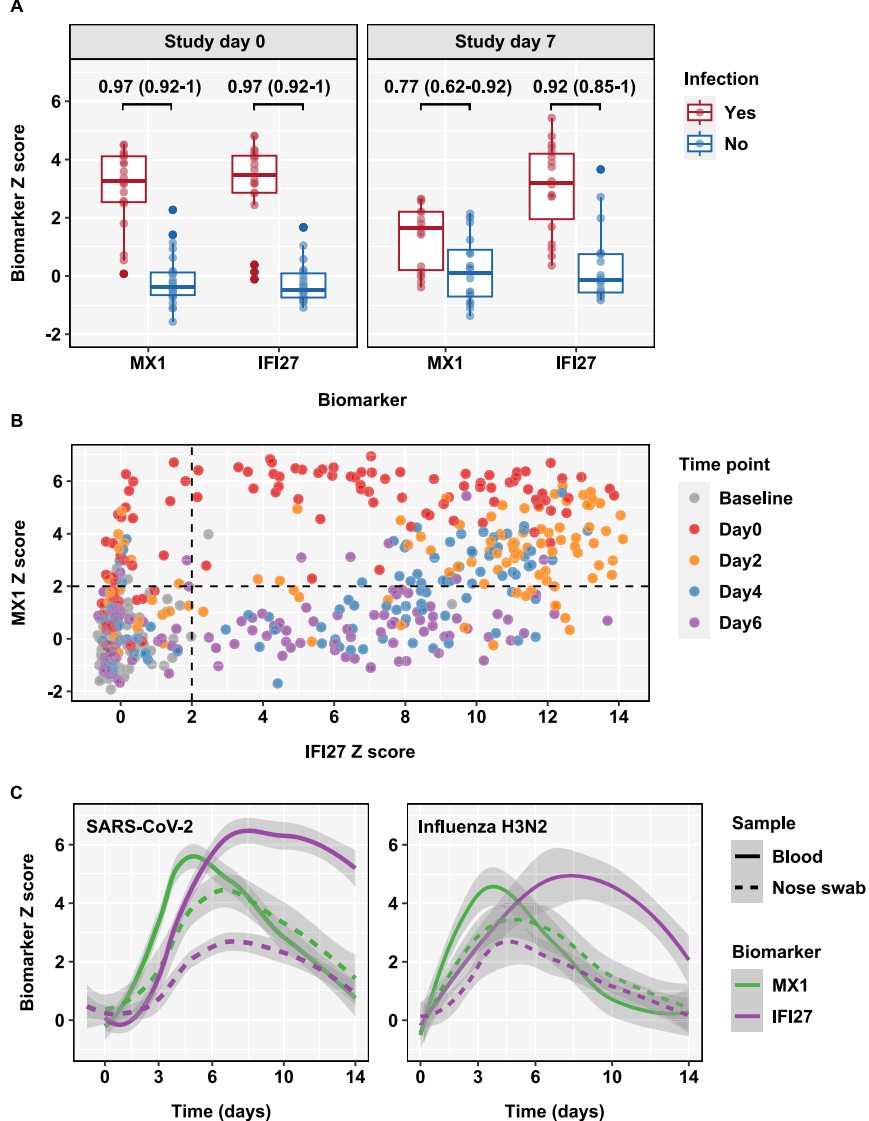

**Fig. 7 | Differences in temporal profiles of blood MX1 and IFI27 expression in naturally acquired respiratory virus infections, and delayed responses in the nose to virus challenge. A** Discrimination between SARS-CoV-2 infected (N = 20) and uninfected (N = 26) household contacts of index cases with COVID-19 by blood transcript levels of *MX1* and *IFI27*, at participant recruitment (study day 0) and 7 days later. Data points represent individual study participants, summarised by box and whisker plots showing median ±interquartile range ± 1.5 ×IQR. Discrimination accuracy is shown as AUROC point estimate and 95% confidence intervals. **B** Individual standardised blood transcript levels of *MX1* against *IFI27* for sequential samples before infection (baseline, N = 128) and at time points indicated (day0, N = 103; day 2, N = 106, day 4, N = 100; day 6, N = 102) after presentation within 48 hours of symptoms onset among prospectively recruited participants with unselected respiratory virus infections. **C** Loess smoothed summary (line ±95% CI) for standardised transcript levels of *MX1* and *IFI27* in blood (N = 17) and nose samples (N = 5-13 for different time points) from participants who developed replicative virus infection by time after SARS-CoV-2 challenge, and in blood (N = 16) and nose samples (N = 12-13 for different time points) from participants who developed replicative virus infection by time after H3N2 influenza challenge.

treatment, and pre-symptomatic screening of contacts of index cases of confirmed viral infections. Notably, for each of these applications, the generalisability of blood transcriptional biomarkers across respiratory viruses may be considered a strength. Translation of these viral biomarkers to near-patient platforms is now required to enable further evaluation of clinical utility and impact in prospective observational and interventional studies.

## Methods
### Research ethics
Regulatory approvals for the human studies presented herein were provided by the UK Health Research Authority under the following reference numbers: 20/UK/2001 and 20/UK/0002 for the SARS-CoV-2 challenge study; 20/NW/0231 for the INSTINCT study; 19/LO/1441 for the H3N2 influenza challenge study. Written informed consent was

provided by all participants directly or by legally authorized representatives for participants under the age of 18.

### Identification of blood transcriptional signatures of viral infection
We updated our previous systematic review of blood transcriptional biomarkers for viral infection[2] (Supplementary Data 1). In the current analysis, we amended our previous eligibility criteria to identify concise blood transcriptional signatures discovered or applied with a primary objective of diagnosis of viral infection from human whole-blood or peripheral blood mononuclear cell samples, excluding those exclusively intended to stratify severity of infection. Other eligibility criteria remained the same as our previous review. In our update, we searched MEDLINE for articles published up to 31 December 2022, using comprehensive MeSH and keyword terms for "viral infection",

"transcriptome", "biomarker", and "blood", as previously[2]. Additional studies were identified in reference lists. Title and abstract screening was independently performed by two reviewers (CT and JGB); short-listed articles were reviewed in full, with input from a third reviewer (RKG) to resolve conflicts. For eligible signatures, constituent genes, modelling approaches and gene weightings were extracted, with verification by a second reviewer. Multi-gene signatures are referred to using a prefix of the first-author's name from the corresponding publication, and a suffix of the number of component genes. Single-gene signatures are referred to by the gene symbol.

## Human challenge and patient cohorts

The SARS-CoV-2 human challenge model has been described previously[4]. Briefly, 36 SARS-CoV-2 unvaccinated seronegative healthy volunteers (age range 18–29 years, 22% female sex, 90% White or Caucasian ancestry) were inoculated intranasally with a standardized dose of D614G-containing pre-alpha wild-type SARS-CoV-2 under quarantine conditions (Fig. 1A). From 24 hours after inoculation, virus was quantified by PCR and culture in samples obtained at 12 hourly intervals from nose (mid-turbinate) and throat swabs for at least 14 days of quarantine, or longer if they remained in quarantine beyond 14 days because they still had detectable virus. A final sample was obtained at 28 days after challenge. Blood samples for RNA sequencing were collected into PAXgene tubes (Qiagen) before virus challenge, 6 hours after challenge, daily thereafter for 14 days and on day 28. Mid-turbinate nose swabs (MW013, MedWire) for RNA sequencing were collected before virus challenge, and on days 1, 3, 5, 7, 10 and 14 after challenge, preserved in RNAprotect (Qiagen). Symptom diaries were collected and viral load was quantified by quantitative RT-PCR using N-gene primers (Forward: GACCCCAAAATCAGCGAAAT, Reverse: TCTGGTTACTGCCAGTTGAATCTG, Probe: ACCCCGCATTACGTTTG GTGGACC), and by culture using focus forming assay (FFA) in Vero cells, as previously described[4]. Three individuals were excluded from the present analysis; one individual who opted out of genetic testing (including RNA sequencing); two individuals who seroconverted in the interval between screening and inoculation were excluded from the present analysis, on the basis that they experienced a recent infection that may affect the biomarker expression that is the focus of this study.

The SARS-CoV-2 household contact study (INSTINCT) has been described previously[39]. Briefly, 52 household contacts (age range 7–79 years, 48% female sex, 90% White ancestry) of SARS-CoV-2 infected index cases recruited within 5 days of index case symptom onset provided nasopharyngeal swabs and blood RNA samples collected in PAXgene tubes on day of enrolment (day 0), day 7, day 14 and day 28. Nasopharyngeal swabs were used to measure viral copy number using PCR against the E-gene (Forward: ACAGGTACGTTAATAGTTAA-TAGCGT, Reverse: ATATTGCAGCAGTACGCACACA; Probe: ACAC-TAGCCATCCTTACTGCGCTTCG-BBQ) as previously described[46].

The Influenza H3N2 human challenge model has been described previously[38]. Briefly, 20 healthy volunteers (age range 22–55 years, 50% female sex, 68% White, 32% Black or Asian ancestry) were inoculated intranasally with a standardized dose of Influenza A/Belgium/4217/2015 (H3N2) under quarantine conditions. From 24 hours after inoculation, virus was quantified by PCR in nasal lavage samples obtained at 12 hourly intervals. Participants were ascertained to have replicative viral infection if found to have consecutive positive PCR tests at least 24 hours after challenge. Blood samples for RNA sequencing (available from 19 participants) were collected into PAXgene tubes before virus challenge and days 1, 2, 3, 7, 10, 14, and 28 after challenge. Nasal curettage samples (available from 17 participants) were collected on days -14 (baseline), 1, 2, 3, 7, 10 and 14 and preserved in TRIzol (ThermoFisher Scientific) as previously described[47].

All RNA samples were stored at −80 °C until processing.

## Transcriptional profiling

Total RNA was extracted from SARS-CoV-2 challenge PAXgene tubes using the PAXgene Blood RNA kit (Qiagen), including on-column DNase treatment and depletion of globin mRNA using the GLOBIN-clear Human Kit (Thermo Fisher Scientific). Total RNA was extracted from the INSTINCT SARS-CoV-2 household contact study and the H3N2 influenza challenge PAXgene tubes using the Qiasymphony PAXgene blood RNA kit (Qiagen), with subsequent DNase I treatment (Zymo) and clean-up using the RNA Clean and Concentrator-96 kit (Zymo), followed by globin mRNA and rRNA depletion using NEBNext® Globin & rRNA Depletion kits (New England BioLabs). Total RNA from SARS-CoV-2 challenge nasopharyngeal swabs and curettage samples was extracted using the RNeasy mini kit (Qiagen), including on-column DNase treatment. RNA concentrations were quantified using Qubit 2.0 Fluorometer (ThermoFisher Scientific). RNA integrity scores were determined using the Bioanalyser (RNA Nano 6000 Chip, Agilent) or 4200 TapeStation (Agilent).

Blood RNA samples from SARS-CoV-2 challenge underwent total RNA sequencing. DNA libraries were constructed using the KAPA RNA HyperPrep Kit with RiboErase (Roche) and sequenced on the Illumina NovaSeq 6000 platform using the NovaSeq 6000 S4 Reagent Kit (200 cycles) (Illumina), giving a median of 69.1 million (range 29.3–152.8) 100 base pair (bp) paired-end reads per sample. Nose swab RNA samples underwent mRNA sequencing. DNA libraries were constructed using the Kappa mRNA HyperPrep kit (Roche) and sequenced on the Illumina NextSeq platform the using the NextSeq 500/550 High Output Kit (75 cycles) (Illumina), giving a median of 32.3 million (range 3.2–176.2) 41 bp paired-end reads per sample. Blood RNA samples from the INSTINCT SARS-CoV-2 household contact study underwent mRNA sequencing. DNA libraries were constructed using the NEB-Next® Ultra™ II Directional RNA Library Prep Kit for Illumina (New England Biolabs) and sequenced on the Illumina HiSeq 4000 using the HiSeq 3000/4000 PE Cluster and SBS kits (Illumina), giving a median of 26.1 million (range 18.34–56.04) 75 bp paired-end reads per sample. Nose curettage RNA samples from the H3N2 human challenge underwent mRNA sequencing. DNA libraries were constructed using the NEBNext® Ultra™ II Directional RNA Library Prep Kit for Illumina (New England Biolabs) and sequenced on the Illumina HiSeq 4000 using the HiSeq 3000/4000 SBS kit (Illumina), giving a median of 74.9 million (range 44.2–122) 75 bp paired-end reads per sample. Whole blood RNA samples from the H3N2 influenza challenge underwent mRNA sequencing, DNA libraries were constructed with the NEBNext® Ultra II Directional RNA Library Prep Kit for Illumina (New England BioLabs) and sequenced on the Illumina NovaSeq 6000 platform using the NovaSeq 6000 S2 200 cycles Flowcell (Illumina), with a target of 40 million paired-end reads per sample.

SARS-CoV-2 challenge sequencing reads were mapped to the reference transcriptome (Ensembl Human GRCh38 release 108) using Kallisto (version 0.46.1)[48]. 360 blood RNA samples from 33 seronegative individuals gave a median of 28.7 million (range 11.6–73.5) mapped reads per sample. 92 nose swab RNA samples gave a median of 23.7 million (range 2.5–138.6) mapped reads per sample. Transcript-level output Deseq2 normalised counts and transcripts per million values were summed on gene level and annotated with Ensembl gene ID, gene name, and gene biotype using the tximport (version 1.20.0) and biomaRt (version 2.48.0) Bioconductor packages in R[49–53].

Sequencing reads from the INSTINCT SARS-CoV-2 household contact study were mapped to the reference transcriptome (NCBI Human GRCh38) using STAR aligner (version 2.7.1a)[54]. 134 blood RNA samples from 52 individuals gave a median of 14.26 million (range 7.30–37.67) mapped reads per sample. Read count matrices were generated using featureCounts from the Rsubread package[55] and normalised using the variance stabilised transformation from the DESeq2 package.

For the sequencing reads of whole blood RNA from the H3N2 influenza challenge, quality control was performed using with FastQC (v 0.11.7; https://www.bioinformatics.babraham.ac.uk/projects/fastqc/) and adapter sequences were removed using Trimmomatic (v 0.36)[56]. The reads were mapped against GRCh38 reference genome using STAR aligner (v 2.7.1a). The featureCounts tool from Subread package (v 1.5.2) was used for transcript quantification. Computed gene counts were used for downstream analyses. Whole blood samples from 19 donors gave a median of 10.1 million (range 8.5–12.7) mapped reads per sample (read length = 100 bp). Transcript-level output Deseq2 normalised counts were annotated with Ensembl gene ID and gene name using biomaRt (version 2.46.3) Bioconductor packages in R.Sequencing reads from the H3N2 challenge were mapped to the reference transcriptome (Ensembl Human GRCh38.p13) using STAR (version 2.7.10a). Nasal RNA samples from 17 donors gave a median of 21.4 million (range 3.32–34.7) mapped reads per sample. Transcript-level output Deseq2 normalised counts were annotated with Ensembl gene ID and gene name using biomaRt (version 2.52.0) Bioconductor packages in R.

For RNAseq datasets that were generated in more than one batch, processing batch effects were excluded by principal component analysis showing that data sets did not cluster separately by sample processing batch (Supplementary Fig. 2). Additional genome-wide transcriptomic microarray data were derived from previously published experimental challenge datasets of other respiratory viruses (GEO accession: GSE73072)[37] and from a natural infection study of respiratory viruses (GEO accession: GSE68310)[40]. In each case, we used log-2 transformed and normalised data matrices to quantify biomarker scores, standardised to baseline samples.

## Signature scores

Analyses were performed in R (version 4.2.2). Biomarker levels were represented by expression values for single genes. Multi-gene transcriptional signatures were calculated as per the original author's descriptions using transcripts per million values, as previously. In one example (Steinbrink19), 3 of 19 genes in the original signature were not available in our RNA sequencing data set, but we included this signature on the basis that the signature score excluding the missing genes still achieved good discrimination of study participants with and without replicative infection across the time course (Supplementary Fig. 3-4). Scores were standardised to Z scores by subtracting the mean and dividing by the standard deviation of pre-inoculation samples, and were multiplied by -1 for scores intended to decrease in the presence of viral infection. Discrimination of each signature for the outcome of replicative infection was calculated as the area under the receiver operating characteristic curve (AUROC), with 95% confidence intervals, and stratified by day since inoculation, using the pROC R package[57]. Correlation between signatures and with viral loads was quantified as Spearman rank correlation coefficients using the ggpubr R package. Graphs were plotted using the ggplot2 R package.

## Analysis of ATACseq data

Publicly available ATAC (Assay for Transposase Accessible Chromatin) sequencing fastq datasets derived from unstimulated human monocytes, B-cells and CD4 T-effector cells (GEO accession: GSE118189, European Nucleotide Archive accession: PRJNA484801)[35] were analysed with the nf-core ATAC-seq analysis pipeline (v2.0) curated in Nextflow[58,59], using default parameters. Adaptors were trimmed using trimgalore (v0.6.7) and reads were aligned to the reference genome (NCBI GRCh38) using BWA (v 0.7.17)[60]. Duplicate reads were identified using picard (v2.27.4)[61]. Reads were filtered using SAMtools (v1.16.1)[62]. BEDtools (v.2.30.0)[63] was used to remove duplicates, reads mapping to blacklisted regions and mitochondrial DNA, multimappers, unmapped reads or those not marked as primary alignments. Replicate datasets were merged using picard for some downstream analyses. Normalised scaled bigWig files were created using BEDtools and tracks were visualised using Integrative Genomics Viewer (v2.16.0)[64]. Peak calling was performed using MACS2 (v2.2.7.1)[65] in broadpeak mode. Peaks were annotated to gene features using HOMER (v4.11)[66] and a consensus peak-set was generated using BEDtools. Matrices of reads falling within consensus peaks were generated using featureCounts from the subread package (v2.0.1)[35] for quantitation.

Publicly available single-cell ATACseq data from the COMBAT consortium[36] (EGAD00001007963; Zenodo: https://doi.org/10.5281/zenodo.6120249) were reanalysed for read counts per cell type in established COVID infection from hospitalised COVID patients. Data were processed as described in the original publication using the ArchR software package (v0.9.3)[67]. The sequencing reads at the *IFI27* locus were plotted per cell type with the plotBrowserTrack function.

## Reporting summary

Further information on research design is available in the Nature Portfolio Reporting Summary linked to this article.

## Data availability

All source data for the analyses presented in this study are provided in the source data file. Processed RNAseq data is available at ArrayExpress for the SARS-CoV-2 challenge study (accession number: E-MTAB-12993, https://www.ebi.ac.uk/biostudies/arrayexpress/studies/E-MTAB-12993). To comply with data privacy restrictions, raw sequencing data is available under managed access through the European Genome-Phenome Archive (https://ega-archive.org), under the following accession numbers: EGAD50000000942 for SARS-CoV-2 challenge study (https://ega-archive.org/datasets/EGAD50000000942), EGAD50000000956 for Influenza H3N2 challenge study (https://ega-archive.org/datasets/EGAD50000000956), and EGAD50000000684 for the INSTINCT SARS-CoV-2 household contact study https://ega-archive.org/datasets/EGAD50000000684). Data will be shared with investigators whose proposed use is within the scope of participant consent subject to a data access agreement. RNAseq data are also available from Gene Expression Omnibus (GEO) under the following accession numbers: GSE73072 for previous human challenge studies, and GSE68310 for previous community acquired respiratory virus infections. ATACseq data are available from GEO under accession number GSE118189 for unstimulated PBMC, and the European Phenome Genome Archive under accession number EGAD00001007963 (https://ega-archive.org/datasets/EGAD00001007931) for the COMBAT consortium data. Source data are provided with this paper.

## Code availability

Custom code for deriving RNA signature Z scores from a gene expression matrix is available on Github: https://github.com/JRosenheim/SARS-CoV-2_challenge/blob/main/Viral_biomarker_scores.R (https://doi.org/10.5281/zenodo.10021757).

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

## Acknowledgements

This research was supported by the Wellcome Trust (224530/Z/21/Z). AL acknowledges funding by the NIHR Health Protection Research Units (HPRU) in Respiratory Infections (NIHR200927). BMC acknowledges funding by the Rosetrees Foundation. CMB, MK and ML acknowledge funding by NIHR Biomedical Research to Imperial College London. CMW acknowledges funding from the Medical Research Council (MR/T016329/1). CT acknowledges funding from the Wellcome Trust (102186/B/13/Z). JCK acknowledges funding from NIHR Oxford Biomedical Research Centre. LCKB acknowledges funding from the NIHR (Academic Clinical Fellowship Programme). LMD acknowledges funding from the European Union's Horizon 2020 research and innovation programme under the Marie Skłodowska-Curie grant agreement No 955321. M.Z.N. acknowledges funding from a MRC Clinician Scientist Fellowship (MR/W00111X/1), Action Medical Research (GN2911) and funding from the Rutherford Fund Fellowship allocated by the MRC UK Regenerative Medicine Platform 2 (MR/5005579/1). MN acknowledges funding from the Wellcome Trust (207511/Z/17/Z) and by NIHR Biomedical Research Funding to UCL and UCLH. R.H. is a NIHR Senior Investigator. RKG acknowledges funding from the National Institute for Health Research (NIHR302829).

## Author contributions

J.R., R.K.G., and M.N. conceived this study. R.H., C.C., and M.N. obtained funding for this study. M.K., A.J.M., A.C., B.K., W.B., and C.C. conceived, obtained funding, and supervised the conduct of the SARS-CoV-2 challenge study. C.T., L.K.B., R.K.G., J.G.B., and M.N. undertook the systematic review of blood transcriptional biomarkers of viral infection. J.R., R.K.G., L.B., and M.N. undertook an analysis of publicly available bulk RNAseq and ATACseq data. J.R., C.T., T.M., L.K.B., J.G.B., H.W., B.L., C.W., C.V., B.M.C., and R.H. undertook sample processing and data analysis for the SARS-CoV-2 challenge study. R.L., L.D., M.Z.N., and S.T. contributed single-cell RNAseq data from the SARS-CoV-2 challenge study. C.M.B., L.P., P.D., M.K., M.L., and C.C. contributed bulk data from the H3N2 influenza challenge study. K.M., E.C., J.F., S.H., and A.L. contributed data from the INSTINCT SARS-CoV-2 household contact study. A.J.K. and J.C.K. contributed data single cell ATACseq data from the COMBAT study. J.R., R.K.G., and M.N. wrote the manuscript with input from all the authors.

## Competing interests

The Authors declare the following competing interests: In the past 3 years, S.A.T. has received remuneration for scientific advisory board membership from Sanofi, GlaxoSmithKline, Foresite Labs and Qiagen. S.A.T. is a co-founder and holds equity in Transition Bio and Ensocell. From 8 January 2024, S.A.T. has been a part-time employee of GlaxoSmithKline. A.J.M., A.C., M.K., M.M. and A.B. are full time employees at hVIVO Services Ltd. No other authors report any competing interests.

## Additional information

Joshua Rosenheim [1], Rishi K. Gupta [2,3], Clare Thakker[1], Tiffeney Mann[1], Lucy C. K. Bell [1], Claire M. Broderick [4], Kieran Madon[5], Loukas Papargyris [4], Pete Dayananda[4], Andrew J. Kwok [6,7], James Greenan-Barrett[3], Helen R. Wagstaffe [4], Emily Conibear [5], Joe Fenn [5], Seran Hakki [5], Rik G. H. Lindeboom[8], Lisa M. Dratva [9,10], Briac Lemetais[1], Caroline M. Weight [1], Cristina Venturini[11], Myrsini Kaforou [4], Michael Levin [4], Mariya Kalinova[12], Alex J. Mann [12], Andrew Catchpole [12], Julian C. Knight [6], Marko Z. Nikolić [3,13], Sarah A. Teichmann [9,10], Ben Killingley[14], Wendy Barclay [4], Benjamin M. Chain [1], Ajit Lalvani[5], Robert S. Heyderman [1], Christopher Chiu [4] & Mahdad Noursadeghi [1] ✉

[1]Division of Infection and Immunity, University College London, London, UK. [2]Institute of Health Informatics, University College London, London, UK. [3]UCL Respiratory, Division of Medicine, University College London, London, UK. [4]Department of Infectious Disease, Imperial College London, London, UK. [5]NIHR Health Protection Research Unit in Respiratory Infections, National Heart and Lung Institute, Imperial College London, London, UK. [6]Centre for Human Genetics, Nuffield Department of Medicine, University of Oxford, Oxford, UK. [7]Department of Medicine and Therapeutics, Faculty of Medicine, The Chinese University of Hong Kong, Shatin, Hong Kong. [8]Wellcome Sanger Institute, Wellcome Genome Campus, Cambridge, UK. [9]Cambridge Stem Cell Institute, University of Cambridge, Cambridge, UK. [10]Department of Medicine, University of Cambridge, Cambridge, UK. [11]Infection, Immunity and Inflammation Department, Great Ormond Street Institute of Child Health, University College London, London, UK. [12]hVIVO Services Ltd, London, UK. [13]Department of Respiratory Medicine, University College London Hospitals NHS Foundation Trust, London, UK. [14]Department of Infectious Diseases, University College London Hospital NHS Foundation Trust, London, UK. ✉e-mail: m.noursadeghi@ucl.ac.uk

