## [Peer Review File · Nature Communications]

Reviewers' Comments:

Reviewer #1:

Remarks to the Author:

SUMMARY

Rosenheim et al. combine a human challenge model approach with blood transcriptomics analysis to determine the performance of blood biomarkers to detect viral infections. Most of the analysis is based on a previously published SARS-CoV-2 human challenge study where 36 healthy adult individuals were exposed to SARS-CoV-2 in a controlled setting. 18 participants developed a replicative infection. Blood samples were collected in PaxGene tubes before and at several time points post-inoculation. Transcriptional changes were determined by RNA-seq. Viral load was determined in nose and throat swabs following PCR or viral culture. Using this dataset, the authors test a variety of published biomarkers for their ability to distinguish replicative from non-replicative infection at different post-infection time points.

The key finding from this analysis is that MX1 and IFI27, two single-gene biomarkers, excel at separating productively from non-productively infected individuals at distinct phases of infection. While MX1 identifies infected individuals early during infection, IFI27 identifies infected individuals during the late phase of infection. The authors validate this temporal behavior in additional viral diseases by using additional publicly available datasets from human challenge models and natural infection.

Developing biomarkers to identify distinct phases of viral infection is important as this increased temporal resolution allows for clinical applications adjusted to specific infection stages (triage, antibiotics subscriptions, antiviral treatment decisions). The study is well performed, and the conclusions are supported by the reported data. However, the study in its current form might not offer novelty of great magnitude as both MX1 and IFI27 were previously described as biomarkers for viral infection (PMID: 7510764; 28619954) and, in the case of MX1, the expression kinetics following infection were already published (PMID: 7510764).

MAJOR POINTS

- The reported biomarkers are based on the type I IFN response. Previous studies have shown that the type I IFN response during viral infections can differ with age, especially between children and adults (PMID: 34408314, 35177862, <https://doi.org/10.1101/2023.01.28.23285133>).
- Furthermore, sex-specific differences were reported in the immune response to COVID-19 (PMID: 36599369). Is it possible to analyze the impact of age and sex on the performance of MX1 and IFI27 as biomarkers?
- It would be important for a viral disease biomarker to distinguish between mild and severe disease. Is it possible to determine whether there are differences in the expression of MX1 and IFI27 based on disease severity as shown for other biomarkers or gene scores (PMID: 33765435)?
- Some SARS-CoV-2-infected individuals have autoantibodies against type I IFNs (PMID: 32972996). Amongst patients with life-threatening COVID-19, the fraction of individuals with these antibodies is estimated to be as high as 10%. The authors should discuss how the occurrence of these autoantibodies would affect the performance and the clinical usage of a type I IFN-based biomarker.
- The authors mention that MX1 and IFI27 could be used in a combined clinical test. Would such a test, based on MX1 and IFI27 levels, be able to estimate how many days have passed between sampling and the infection event?

MINOR POINTS

- All Figures: the authors should pick a specific color code for MX1 and IFI27 (e.g., red and blue) and a distinct color code for replicative and non-replicative infection (e.g., green and purple). The current setting with red and blue being used in both cases is confusing.
- Figure 1: there is a typo in the caption "replicative virus replication."
- Figure 1: could the authors add labels to the heat map clusters stating "replicative infection" and "non-replicative infection."
- Figure 3b: The authors should plot nose and throat samples in separate panels. The current representation makes it difficult to discern any trend.
- Figure 3c: What are the authors' thoughts on the fact that MX1 correlates well with viral load,

although the viral kinetics seem better aligned with IFI27 expression?

- Figure 7c: these plots are crowded. It might be easier to spot the key messages if there were separate plots for MX1 and IFI27

Reviewer #2:

Remarks to the Author:

The paper by Rosenheim and colleagues uses data from human challenge models to assess the kinetics of the host-response blood transcriptome following viral infection. The authors show that MX1 transcripts have good diagnostic accuracy in early infection, correlation with viral load and identification of virus culture positivity. In contrast, IFI27 showed a delayed response which was restricted to myeloid peripheral blood cells. Compared to the blood, MX1 and IFI27 expression in the nose was less sensitive and did not discriminate between early and late phases of infection.

Overall, this is a well written paper and makes good use of a broad range of data derived from human challenge models with different pathogens. I have the following questions for the authors

- Line 187-189 Where component genes could not be identified in the RNA seq dataset (for example due to genes being withdrawn from reference transcriptome), these genes were excluded from calculations.
 - o This process of removing genes from signatures - depending on the method by which the signature is generated/ works to discriminate conditions - could be impacted by adhoc removing genes.
 - o More clarity as to how frequently genes were removed from signatures, how many genes were removed and for exactly why genes were removed is needed.
 - o In addition, access to code and scores for each of the individual transcriptome signatures derived from the meta analysis should be provided.
- A threshold of $Z > 2$ appears to be used frequently for both single gene transcripts of MX1 and IFI27 - however no context is given as to how this threshold is determined. Clarity in the methods section as to how and WHY this threshold was generated is required
- More clarity is required regarding how single gene transcript signatures are applied to the data. E.g. we have information that describes how this single gene was derived but no explanation on how this gene predicts infection? Is it just looking to see if its different from the average of the non-replicative individuals? Or different from the mean of the baseline samples taken?
- The low signal in the nose is interesting - do you think that this is a biological phenomenon or simply reflective of the fact that a specialised samples/processing approach is needed for host RNA in the nose. What do the authors predict triggers the IFN response in the blood if it is not coming from an IFN signature in the nose?

Minor Issues

- Line 107 states that quantification of viral load and symptoms has been described previously (with reference 4). Given that your other cohorts are described with inclusion of which primers were used in the assay - this may be an important consideration.
 - o The human challenge study quantified viral load using N gene primers, whereas the E gene is used for the natural challenge samples collected. Given in some analyses samples are normalised to PCR viral load - it is important for readers to understand how viral load is quantified in both cases. This information should be consistently available for all cohorts.
 - Clarify regarding the reference genomes (could you identify them all in the same syntax) so we can understand how comparable they are.
 - o Hg38 and GRCh38 are the same - release 108 vs .p13 are different versions. This could be made more clear by referencing them all in the same format.
 - Batch effects removed via PCR of all RNAseq data (supplementary fig2) - could you please add additional comments to the methods to show reasoning for ruling out batch effects from data.
 - Viral culture - method should be referenced. Reference viral load and symptom quantification (line 107)
 - Figure 1 A/b (what is happening with the first subject - they're positive on day 28?). Additionally, what method is used to cluster these individuals - perhaps stated in the caption?
 - Figure 2 - figure 2C is mislabelled as 2B (this appears twice)
 - Figure 6 caption - line 678 (state n=16 replicative and n=3 non-replicative - what happened to

the 17th individual who registered as replicative in the original paper?). This is also evident in the methods where you state that 20 individuals were infected - but then only 19 blood samples.

Reviewer #3:

Remarks to the Author:

Comments

In this study, Rosenheim and colleagues performed a prospective controlled human model of SARS-CoV-2 infection, complemented with longitudinal sampling to measure viral replication and host responses covering the full time course of viral replication using RNA sequencing data from the blood of SARS-CoV-2 positive individuals. The main objective of the study was to investigate differential temporal profiles of MX1 and IFI27 responses from the SARS-CoV-2 challenge model. The findings indicate that blood-based MX1 and IFI27 serve as more effective transcriptional biomarkers for early and late phases of SARS-CoV-2 and Influenza H3N2 human challenges when compared to nasal mucosal samples.

It's important to note that the expression of MX1 and IFI27 in respiratory viral infections is not a novel finding, as evidenced by published papers. For instance, Bizzotto and colleagues demonstrated a high correlation between MX1 expression and viral load (<https://doi.org/10.1016/j.isci.2020.101585>). Additionally, several published papers have shown the overexpression of IFI27, as indicated in the reference section. In this study, measurements were conducted within a controlled human model, which holds valuable insights into the molecular mechanisms underlying the pathogenesis of SARS-CoV-2 and H3N2 influenza infections. However, I have doubts about how applicable these findings might be in the context of real-world infections. However, should it be considered for publication, the following points need to be addressed or clarified further by the authors.

1. Methods should be described in sufficient detail for interested researchers to be able to reproduce the procedure. Supplier (city, state, country) for all reagents, kits, and equipment used need to be provided.
2. It would be more helpful to have a schematic summary showing both study design and patient cohort.
3. The method for mapping the RNASeq data to gene counts is not described and the normalisation of gene counts to a housekeeping reference gene is not mentioned. Important QC information re batch effects and the need for batch correction need further clarification.
4. Figure 7A (and not 7B) effectively displays MX1 and IFI27 "Z scores" for d0 and d7. It would be advantageous to have this format, similar to 7A, for other time points (potentially in the supplement).
5. The key characteristics of the study participants should be shown in a demographic table to see the generalisability of the findings.
6. Though both MX1 and IFI27 are involved in the innate immune response, have the authors assessed post-vaccine (any vaccine) adaptive immune responses that could potentially influence this study?
7. Would the author able to graph expression trend (asymptomatic, symptomatic and recovery) during the recovery period?
8. Were blood samples collected in different types of tubes (e.g., EDTA) for the protein measurements of MX1 and IFI27 and possibly cytokines measurement?
9. This study is presently limited to analysing gene expression profiles with the pre-alpha wild-type SARS-CoV-2 and H3N2 strains. However, to enhance patient management, it's essential to expand testing to cover other available virus variants. It's worth noting that, in certain instances, due to the virus's lethal nature, this might not be feasible.

In summary, the data presented in the current manuscript does not appear to align with the authors' intended goal for translational applications.

Monday, 25 September 2023

Re: NCOMMS-23-33327-T - SARS-CoV-2 human challenge reveals single-gene blood transcriptional biomarkers that discriminate early and late phases of acute respiratory viral infections.

Response to reviewers

Reviewer 1

- 1. Rosenheim et al. combine a human challenge model approach with blood transcriptomics analysis to determine the performance of blood biomarkers to detect viral infections. Most of the analysis is based on a previously published SARS-CoV-2 human challenge study where 36 healthy adult individuals were exposed to SARS-CoV-2 in a controlled setting. 18 participants developed a replicative infection. Blood samples were collected in PaxGene tubes before and at several time points post-inoculation. Transcriptional changes were determined by RNA-seq. Viral load was determined in nose and throat swabs following PCR or viral culture. Using this dataset, the authors test a variety of published biomarkers for their ability to distinguish replicative from non-replicative infection at different post-infection time points. The key finding from this analysis is that MX1 and IFI27, two single-gene biomarkers, excel at separating productively from non-productively infected individuals at distinct phases of infection. While MX1 identifies infected individuals early during infection, IFI27 identifies infected individuals during the late phase of infection. The authors validate this temporal behavior in additional viral diseases by using additional publicly available datasets from human challenge models and natural infection. Developing biomarkers to identify distinct phases of viral infection is important as this increased temporal resolution allows for clinical applications adjusted to specific infection stages (triage, antibiotics subscriptions, antiviral treatment decisions). The study is well performed, and the conclusions are supported by the reported data. However, the study in its current form might not offer novelty of great magnitude as both MX1 and IFI27 were previously described as biomarkers for viral infection (PMID: 7510764; 28619954) and, in the case of MX1, the expression kinetics following infection were already published (PMID: 7510764).*

We thank the reviewer for their useful summary of the study design, main findings and supportive remarks on the technical quality of the study and the importance of our findings. The fact that MX1 and IFI27 have been previously reported as biomarkers of viral infection is an explicit component of the study design which focuses on previously reported biomarkers of viral infection. The key conceptual advance is that different biomarkers discriminate early and late phases of infection- a short-lived MX1 response in the hyperacute phase, and a delayed but sustained IFI27 response thereafter. This finding has two important translational applications that we show are generalisable to common respiratory viruses:

- The SARS-CoV-2 challenge experiment revealed this biology, but validation in the longitudinal observational cohorts of community acquired infection (Figure 7A and 7B) show that IFI27 achieves greater diagnostic accuracy than MX1 because of the delay between transmission and testing in naturally acquired infection.
- MX1 correlates best with viral load (Figure 3B, Supp Figure 6) and predicts infectiousness (Figure 4), providing the first evidence for a biomarker that can be used to identify patients most likely to benefit from antiviral treatments and infection control measures (such as quarantine/self-isolation).

These have the potential to transform diagnostic yield for respiratory virus infections and to limit antivirals and infection control interventions to those who actually need them. Both invaluable tools for tackling endemic respiratory viruses and our response to the next pandemic.

Additional important new insights are that we make significant inroads into the mechanism for the differences in the temporal profiles of each biomarker through cell-type specific epigenetic regulation (Figure 5, Supp Figure 7), and intriguingly, show that there is greater utility in measuring these responses in blood than at the primary site of infection in the nose (Figure 7C).

We hope we have appropriately highlighted these advances in the abstract and discussion of the manuscript.

2. *The reported biomarkers are based on the type I IFN response. Previous studies have shown that the type I IFN response during viral infections can differ with age, especially between children and adults (PMID: 34408314, 35177862, <https://doi.org/10.1101/2023.01.28.23285133>). Furthermore, sex-specific differences were reported in the immune response to COVID-19 (PMID: 36599369). Is it possible to analyze the impact of age and sex on the performance of MX1 and IFI27 as biomarkers?*

We thank the reviewer for an important question. The age range in this study was limited to young adults to enable direct informed consent and to mitigate against the risk of severe COVID19 associated with advancing age. The analysis of MX1 and IFI27 in the INSTINCT cohort of community acquired SARS-CoV-2 infection included participants from 7-77 years of age, providing evidence generalisability across a wider age range. Both sexes are included in the SARS-CoV-2 human challenge study We have added information on the age range in each cohort to the methods text (lines 100, 116 and 123). Therefore, we extended the analysis to compare the temporal profile of MX1 and IFI27 transcripts in blood by sex. We found no between group differences by sex. We have included this analysis as an additional panel in Supplementary Figure 4 and in the results narrative (lines 260-262).

3. *It would be important for a viral disease biomarker to distinguish between mild and severe disease. Is it possible to determine whether there are differences in the expression of MX1 and IFI27 based on disease severity as shown for other biomarkers or gene scores (PMID: 33765435)?*

We acknowledge the interest in biomarkers for severity of disease, or to predict progression of disease severity. This was beyond the scope of the present study design because existing respiratory virus human challenge models were specifically designed to minimise risk of severe disease. As described in the published clinical characterisation of the SARS-CoV-2 model (*Nat Med* 2022; **28**: 1031–1041. <https://doi.org/10.1038/s41591-022-01780-9>), the spectrum of clinical symptoms experienced by the participants was limited to mild disease only. We discuss the fact that our conclusions are currently limited to data derived from individuals with non-severe infection, and the need for future validation in hospitalised cohorts for whom the time of exposure can be estimated will be required to assess whether severe disease alters the temporal profiles of these biomarkers. (lines 440-442).

4. *Some SARS-CoV-2-infected individuals have autoantibodies against type I IFNs (PMID: 32972996). Amongst patients with life-threatening COVID-19, the fraction of individuals with these antibodies is estimated to be as high as 10%. The authors should discuss how the occurrence of these autoantibodies would affect the performance and the clinical usage of a type I IFN-based biomarker.*

We thank the reviewer for an important question. We have revised the discussion to highlight the possibility that anti-type 1 IFN antibodies may reduce the expression of IFN-inducible genes and therefore the sensitivity of these biomarkers (lines 442-445).

5. *The authors mention that MX1 and IFI27 could be used in a combined clinical test. Would such a test, based on MX1 and IFI27 levels, be able to estimate how many days have passed between sampling and the infection event?*

The reviewer raises an interesting question. We undertook an additional analysis to address this point when at least one biomarker was elevated (Z score>2) indicative of an infection. Both biomarkers showed statistically significant linear relationships with time from challenge in opposing directions. The correlation coefficient for MX1 was greater than that of IFI27, but the ratio of the two measurements (MX1:IFI27) further improved the strength of the correlation with time, and transitioned from a positive value to a negative value at day 5. We have included this analysis in the results narrative (lines 287-290) and in a new Supplementary Figure 5.

6. *All Figures: the authors should pick a specific color code for MX1 and IFI27 (e.g., red and blue) and a distinct color code for replicative and non-replicative infection (e.g., green and purple). The current setting with red and blue being used in both cases is confusing.*

We acknowledge the reviewer's request and have amended the figures as suggested in Figures 4B, 6A, and 7C.

7. *Figure 1: there is a typo in the caption "replicative virus replication."*

We have corrected this error (line 669).

8. *Figure 1: could the authors add labels to the heat map clusters stating "replicative infection" and "non-replicative infection."*

We have amended the figure as suggested.

9. *Figure 3b: The authors should plot nose and throat samples in separate panels. The current representation makes it difficult to discern any trend.*

We acknowledge the reviewer's request. We highlight the relationship between viral load from nose and throat swabs with each biomarker is shown in separate plots in Supplementary Figure 6. We are happy to accept editorial advice on whether to divide the data in Figure 3B into separate plots also, but prefer the current arrangement for efficient use of display panels in the main manuscript.

10. *Figure 3c: What are the authors' thoughts on the fact that MX1 correlates well with viral load, although the viral kinetics seem better aligned with IFI27 expression?*

The reviewer highlights an interesting observation referring to Figure 3C. The lower correlation of viral load with IFI27 is the result of measurements at late time points in which IFI27 levels remain elevated despite the decay in viral load. We have amended the results narrative to highlight this point (lines 295-296).

11. *Figure 7c: these plots are crowded. It might be easier to spot the key messages if there were separate plots for MX1 and IFI27*

We are happy to accept editorial advice on this point, but our preference is to retain the current format of this figure to highlight that differential temporal profiles of MX1 and IFI27 levels in blood are associated with earlier responses in the blood than the nose.

Reviewer 2

12. *The paper by Rosenheim and colleagues uses data from human challenge models to assess the kinetics of the host-response blood transcriptome following viral infection. The authors show that MX1 transcripts have good diagnostic accuracy in early infection, correlation with viral load and identification of virus culture positivity. In contrast, IFI27 showed a delayed response which was restricted to myeloid peripheral blood cells. Compared to the blood, MX1 and IFI27 expression in the nose was less sensitive and did not discriminate between early and late phases of infection. Overall, this is a well written paper and makes good use of a broad range of data derived from human challenge models with different pathogens. I have the following questions for the authors*

We are grateful for the reviewer's supportive comments.

13. *Line 187-189 Where component genes could not be identified in the RNA seq dataset (for example due to genes being withdrawn from reference transcriptome), these genes were excluded from calculations.*

a. *This process of removing genes from signatures - depending on the method by which the signature is generated/ works to discriminate conditions - could be impacted by adhoc removing genes. More clarity as to how frequently genes were removed from signatures, how many genes were removed and for exactly why genes were removed is needed.*

We acknowledge the reviewer's request for greater clarity. Missing genes were limited to one signature only- Steinbrink19, in which 3 of 19 genes in the original signature were not available in our RNA sequencing data set. Nonetheless, we included this signature on the basis that signature score excluding the missing genes still achieved good discrimination of study participants with and without replicative infection across the time course (Supplementary Figure 3-4). We have amended the methods text (lines 196-199) and footnotes to Supplementary Table 1 to provide this detail.

b. *In addition, access to code and scores for each of the individual transcriptome signatures derived from the meta analysis should be provided.*

We have added a link to the code for this analysis on Github (https://github.com/JRosenheim/SARS-CoV-2_challenge/blob/main/Viral_biomarker_scores.R) and an additional supplementary file of all signature scores for study subjects by time point and allocation to replicative and non-replicative infection group, as well supporting data for all other challenge and community infection cohorts to allow enable readers to recreate all the biomarker analyses presented in the manuscript. The data availability statement has updated accordingly (lines 482-484).

14. *A threshold of Z>2 appears to be used frequently for both single gene transcripts of MX1 and IFI27 - however no context is given as to how this threshold is determined. Clarity in the methods section as to how and WHY this threshold was generated is required.*

Z scores were standardised to the distribution of measurements in baseline (pre-inoculation) samples to represent a normal distribution in the absence of infection. In this context, we used a pre-specified threshold

of Z score >2 provides to indicate increased biomarker levels compared to baseline with ~98% specificity. We have amended the results narrative to provide this rationale as requested (lines 267-268).

15. More clarity is required regarding how single gene transcript signatures are applied to the data. E.g. we have information that describes how this single gene was derived but no explanation on how this gene predicts infection? Is it just looking to see if it's different from the average of the non-replicative individuals? Or different from the mean of the baseline samples taken?

For single genes, the expression level (Log2 transformed Deseq2 normalised gene counts or transcripts per million) was used and standardised against the distribution of gene expression values for that gene in baseline/pre-inoculation samples. We have amended the methods text to provide this detail (lines 194-195).

16. The low signal in the nose is interesting – do you think that this is a biological phenomenon or simply reflective of the fact that a specialised samples/processing approach is needed for host RNA in the nose. What do the authors predict triggers the IFN response in the blood if it is not coming from an IFN signature in the nose?

The reviewer raises an important question. We do expect that the type 1 IFN response is initiated at the site of replicative viral infection. The fact that IFI27 expression in blood is largely restricted to myeloid cells is consistent with the hypothesis that quantitatively lower Z scores for IFI27 in nasal swabs may reflect fewer immune cells in superficial nasal samples, since these cells may reside primarily in sub-epithelial layers. MX1 expression appears to be less restricted to specific cell types. Therefore, lower MX1 Z scores suggest that nasal swabs are less sensitive because the sample provides lower sequencing read depth or the cells captured by nasal swabs do not upregulate IFN stimulated gene expression to the same extent as circulating immune cells. We have amended the discussion text (lines 412-418) to further clarify our inferences.

Line 107 states that quantification of viral load and symptoms has been described previously (with reference 4). Given that your other cohorts are described with inclusion of which primers were used in the assay - this may be an important consideration. The human challenge study quantified viral load using N gene primers, whereas the E gene is used for the natural challenge samples collected. Given in some analyses samples are normalised to PCR viral load - it is important for readers to understand how viral load is quantified in both cases. This information should be consistently available for all cohorts.

We have amended the methods to provide these details as requested (lines 109-112).

17. Clarify regarding the reference genomes (could you identify them all in the same syntax) so we can understand how comparable they are. Hg38 and GRCh38 are the same - release 108 vs .p13 are different versions. This could be made more clear by referencing them all in the same format.

We have amended the text as requested (line 176).

18. •Batch effects removed via PCR of all RNAseq data (supplementary fig2) - could you please add additional comments to the methods to show reasoning for ruling out batch effects from data.

Batch effects are excluded by the fact that data sets do not cluster separately by sample processing batch. We have amended the methods text to make this clearer (line 187).

19. Viral culture - method should be referenced. Reference viral load and symptom quantification (line 107)

We have updated the methods section to provide the additional detail as requested (lines 109-112).

20. Figure 1 A/b (what is happening with the first subject - they're positive on day 28?). Additionally, what method is used to cluster these individuals - perhaps stated in the caption?

This subject developed community acquired SARS-CoV-2 infection after leaving quarantine on day 14. This was captured incidentally by nose and throat PCRs on prespecified day 28 follow up time point. Complete linkage hierarchical clustering was used. We have amended the figure legend to provide this detail (lines 664-668).

21. Figure 2 - figure 2C is mislabelled as 2B (this appears twice)

We have corrected this error.

22. *Figure 6 caption - line 678 (state n=16 replicative and n=3 non-replicative - what happened to the 17th individual who registered as replicative in the original paper?). This is also evident in the methods where you state that 20 individuals were infected - but then only 19 blood samples.*

We thank the reviewer for highlighting this discrepancy. We have amended the methods text to clarify that blood RNA samples were available from 19 participants and nose RNA samples from 17 participants for host transcriptional profiling (lines 127-129).

Reviewer 3

23. *In this study, Rosenheim and colleagues performed a prospective controlled human model of SARS-CoV-2 infection, complemented with longitudinal sampling to measure viral replication and host responses covering the full time course of viral replication using RNA sequencing data from the blood of SARS-CoV-2 positive individuals. The main objective of the study was to investigate differential temporal profiles of MX1 and IFI27 responses from the SARS-CoV-2 challenge model. The findings indicate that blood-based MX1 and IFI27 serve as more effective transcriptional biomarkers for early and late phases of SARS-CoV-2 and Influenza H3N2 human challenges when compared to nasal mucosal samples. It's important to note that the expression of MX1 and IFI27 in respiratory viral infections is not a novel finding, as evidenced by published papers. For instance, Bizzotto and colleagues demonstrated a high correlation between MX1 expression and viral load (<https://doi.org/10.1016/j.isci.2020.101585>). Additionally, several published papers have shown the overexpression of IFI27, as indicated in the reference section. In this study, measurements were conducted within a controlled human model, which holds valuable insights into the molecular mechanisms underlying the pathogenesis of SARS-CoV-2 and H3N2 influenza infections. However, I have doubts about how applicable these findings might be in the context of real-world infections. However, should it be considered for publication, the following points need to be addressed or clarified further by the authors.*

As in our response to comment 1 (reviewer 1 The fact that MX1 and IFI27 have been previously reported as biomarkers of viral infection is an explicit component of the study design which focuses on previously reported biomarkers of viral infection. The key conceptual advance is that different biomarkers discriminate early and late phases of infection- a short-lived MX1 response in the hyperacute phase, and a delayed but sustained IFI27 response thereafter. This finding has two important translational applications that we show are generalisable to common respiratory viruses:

- The SARS-CoV-2 challenge experiment revealed this biology, but validation in the longitudinal observational cohorts of community acquired infection (Figure 7A and 7B) show that IFI27 achieves greater diagnostic accuracy than MX1 because of the delay between transmission and testing in naturally acquired infection.
- MX1 correlates best with viral load (Figure 3B, Supp Figure 6) and predicts infectiousness (Figure 4), providing the first evidence for a biomarker that can be used to identify patients most likely to benefit from antiviral treatments and infection control measures (such as quarantine/self-isolation).

These have the potential to transform diagnostic yield for respiratory virus infections and to limit antivirals and infection control interventions to those who actually need them. Both invaluable tools for tackling endemic respiratory viruses and our response to the next pandemic.

Additional important new insights are that we make significant inroads into the mechanism for the differences in the temporal profiles of each biomarker through cell-type specific epigenetic regulation (Figure 5, Supp Figure 7), and intriguingly, show that there is greater utility in measuring these responses in blood than at the primary site of infection in the nose (Figure 7C).

24. *Methods should be described in sufficient detail for interested researchers to be able to reproduce the procedure. Supplier (city, state, country) for all reagents, kits, and equipment used need to be provided.*

We have provided the manufacturer/supplier details for specific reagents and equipment required to reproduce our experiments, consistent with contemporary practice in Nature Communications. We are happy to provide any additional detail on editorial advice.

25. *It would be more helpful to have a schematic summary showing both study design and patient cohort.*

We had already provided a schematic for the blood sampling framework in the SARS-CoV-2 challenge experiments. In the revised manuscript, we have replaced this with a more comprehensive schematic figure to summarise the timeline for blood, nose and throat sampling for host RNA biomarkers and virus PCRs

(Figure 1A). In addition, we have amended the methods text to provide summary level data on the characteristics of the cohort (lines 99-101).

26. The method for mapping the RNASeq data to gene counts is not described and the normalisation of gene counts to a housekeeping reference gene is not mentioned. Important QC information re batch effects and the need for batch correction need further clarification.

We believe we have provided all the relevant technical detail for mapping RNAseq data to gene counts. We would be happy to provide any specific additional detail requested. The mapped data are output as Deseq2 normalised counts or TPM as specified, and biomarkers measurements are subsequently scaled by Z score standardisation using the distribution of data from baseline/pre-inoculation samples. PCA of genome-wide mapped data excludes clustering by batch (Supplementary Figure 2), hence no further normalisation was required. As in our response to comment 18 (reviewer 2), we have amended the methods text to make this clearer (line 187).

27. Figure 7A (and not 7B) effectively displays MX1 and IFI27 "Z scores" for d0 and d7. It would be advantageous to have this format, similar to 7A, for other time points (potentially in the supplement).

Boxplots of data stratified by sampling day are provided in Supplementary Figure 8. We believe this addresses the reviewer's request.

28. The key characteristics of the study participants should be shown in a demographic table to see the generalisability of the findings.

These characteristics are provided in detail in the cited publications for each cohort. For the readers' convenience, we have provided summary level characteristics for the participants in the SARS-CoV-2 and Influenza H3N2 challenge cohorts within the methods text (lines 99-101, line 116 and line 120).

29. Though both MX1 and IFI27 are involved in the innate immune response, have the authors assessed post-vaccine (any vaccine) adaptive immune responses that could potentially influence this study?

None of the participants in the SARS-CoV-2 human challenge experiment had received SARS-CoV-2 vaccines prior to the challenge. We already state that all participants were seronegative but have amended the methods text to provide this additional clarification (lines 99-100).

30. Would the author able to graph expression trend (asymptomatic, symptomatic and recovery) during the recovery period?

We are not confident that we correctly understand this question. Our best estimation is that the reviewer is asking for an evaluation of MX1 and IFI27 expression levels in relation to symptoms and to the recovery phase among those who develop replicative infection. Figure 3A provides a graphical representation of the relationship between biomarker levels and symptoms stratified by time point. In this analysis, biomarker levels show no correlation with symptoms. Data from late time points with 'zero' symptoms (representative of the recovery phase) show falling MX1 Z scores compared to MX1 levels at earlier time points, and compared to contemporary IFI27 levels that remain elevated. We hope that this addresses the reviewer's question. We apologise if we have misunderstood and happy to respond to any further clarification from the reviewer.

31. Were blood samples collected in different types of tubes (e.g., EDTA) for the protein measurements of MX1 and IFI27 and possibly cytokines measurement?

EDTA samples are not available to us for protein level measurements. Selected cytokines have been measured in serum samples. They are the focus of on-going work and beyond the scope of the objectives of the present study. Importantly, MX1 and IFI27 are both intracellular proteins and cannot be accurately quantified in serum.

32. This study is presently limited to analysing gene expression profiles with the pre-alpha wild-type SARS-CoV-2 and H3N2 strains. However, to enhance patient management, it's essential to expand testing to cover other available virus variants. It's worth noting that, in certain instances, due to the virus's lethal nature, this might not be feasible.

The reviewer correctly highlights that the SARS-CoV-2 challenge data are derived from human challenge experiments with pre-alpha ancestral SARS-CoV-2 virus. Data from challenge studies with other SARS-CoV-2 variants are not yet available. However, we have replicated our findings in all other available acute respiratory virus challenges including multiple influenza strains, rhinovirus and RSV (Figure 6), naturally acquired SARS-CoV-2 virus infection, spanning pre-alpha, and alpha-virus (B.1.1.7) pandemic waves in the UK, and unselected naturally acquired acute respiratory virus infection pre-pandemic. All the available

data suggests that our findings are generalisable to non-severe acute respiratory virus infection caused by diverse virus pathogens. We have amended the results text to highlight the fact that naturally acquired SARS-CoV-2 infection cohort spanned pre-alpha and alpha-virus strains (lines 345-346).

33. In summary, the data presented in the current manuscript does not appear to align with the authors' intended goal for translational applications.

We respectfully disagree with the reviewer's comment. The translational impacts of our study are response to comment 1 and comments 23.

Reviewers' Comments:

Reviewer #1:

Remarks to the Author:

The revised manuscript has been improved. I find the novel results in Supplementary Figure 5 interesting, and I really appreciate the highlighting of the significance in the authors' response to comment 1. I have a few minor comments for the authors to consider:

1. The explanation of the significance in comment 1 is really clear. The authors should add some of these statements in the same concise and clear manner to the summary section and the beginning/end of the discussion. As the authors report novel results on previously published signatures, having these statements highlight the novelty would be really helpful.

2. Figure 3B: In the center of the point cloud, I cannot discern whether dots originate from nasal or throat swabs. If the authors prefer not to split up the plot, they should take other actions to increase clarity (smaller icons, larger plotting area, lines around symbols).

Reviewer #2:

Remarks to the Author:

The authors have addressed all reviewer comments and I believe that this manuscript should proceed to publication.

Reviewer #3:

Remarks to the Author:

Dear authors,

I appreciate your diligence in addressing my suggestions and your commitment to enhancing the overall quality of your work. I believe that the revisions have significantly strengthened the quality and clarity of the manuscript. I have no additional comments to make.

Wednesday, 08 November 2023

Re: NCOMMS-23-33327A - SARS-CoV-2 human challenge reveals single-gene blood transcriptional biomarkers that discriminate early and late phases of acute respiratory viral infections.

Response to reviewers

Reviewer 1

1. *The revised manuscript has been improved. I find the novel results in Supplementary Figure 5 interesting, and I really appreciate the highlighting of the significance in the authors' response to comment 1. I have a few minor comments for the authors to consider:*

We thank the reviewer for their supportive remarks and address their remaining comments below.

2. *The explanation of the significance in comment 1 is really clear. The authors should add some of these statements in the same concise and clear manner to the summary section and the beginning/end of the discussion. As the authors report novel results on previously published signatures, having these statements highlight the novelty would be really helpful.*

We have revised the summary section at the end of the discussion to incorporate the key conceptual advances of our study as suggested by the reviewer (lines 288-297).

3. *Figure 3B: In the centre of the point cloud, I cannot discern whether dots originate from nasal or throat swabs. If the authors prefer not to split up the plot, they should take other actions to increase clarity (smaller icons, larger plotting area, lines around symbols).*

We have revised Fig 3B to make the distinction between nose and throat swab data points clearer as suggested by the reviewer.

Reviewer 2

4. *The authors have addressed all reviewer comments and I believe that this manuscript should proceed to publication.*

We are grateful for the reviewer's supportive comments.

Reviewer 3

5. *Dear authors, I appreciate your diligence in addressing my suggestions and your commitment to enhancing the overall quality of your work. I believe that the revisions have significantly strengthened the quality and clarity of the manuscript. I have no additional comments to make.*

We are grateful for the reviewer's supportive comments.